# Molecular Characteristics, Functional Definitions, and Regulatory Mechanisms for Cross-Presentation Mediated by the Major Histocompatibility Complex: A Comprehensive Review

**DOI:** 10.3390/ijms25010196

**Published:** 2023-12-22

**Authors:** Sen Liu, Shaoqiang Wei, Yan Sun, Guowei Xu, Shidong Zhang, Jianxi Li

**Affiliations:** Engineering Technology Research Center of Traditional Chinese Veterinary Medicine of Gansu Province, Lanzhou Institute of Animal Husbandry and Pharmaceutical Sciences, Chinese Academy of Agricultural Sciences, Lanzhou 730050, China; 82101225776@caas.cn (S.L.); 82101212380@caas.cn (S.W.); sunyan01@126.com (Y.S.); xuguowei@caas.cn (G.X.)

**Keywords:** MHC, cross-presentation, exogenous antigens, immune response

## Abstract

The major histocompatibility complexes of vertebrates play a key role in the immune response. Antigen-presenting cells are loaded on MHC I molecules, which mainly present endogenous antigens; when MHC I presents exogenous antigens, this is called cross-presentation. The discovery of cross-presentation provides an important theoretical basis for the study of exogenous antigens. Cross-presentation is a complex process in which MHC I molecules present antigens to the cell surface to activate CD8^+^ T lymphocytes. The process of cross-representation includes many components, and this article briefly outlines the origins and development of MHC molecules, gene structures, functions, and their classical presentation pathways. The cross-presentation pathways of MHC I molecules, the cell lines that support cross-presentation, and the mechanisms of MHC I molecular transporting are all reviewed. After more than 40 years of research, the specific mechanism of cross-presentation is still unclear. In this paper, we summarize cross-presentation and anticipate the research and development prospects for cross-presentation.

## 1. Introduction

The major histocompatibility complex (MHC) is distributed in all vertebrates and is encoded by the major histocompatibility antigen gene group [1]. The concept of MHC appears to explain the phenomenon of tissue rejection and compatibility between different individuals [2]. With the in-depth study of molecular biology and genetics, the polymorphism of the MHC gene group and its correlation with anti-infection and immunity have been gradually revealed [3,4]. The product encoded by its genetic code participates in the immune response and acts as a crucial component of the immune system. Its primary function is to present antigens to the immune cells [5].

## 2. Discovery of MHC

MHC was first identified in inbred mice. In 1948, George D. Snell was conducting experiments on tumor transplants when he discovered that there were immune rejection reactions between different mice. Then, the antigen responsible for rejection following transplantation was discovered and named the histocompatibility antigen, and H-2 was defined as the MHC in mice. In 1958, the Dausset study discovered the human leukocyte antigen HLA and the HLA gene, which is equivalent to the mouse H gene. In 1963, when studying the phenomenon of organ transplant rejection, Baruj Benacerraf observed that the antibody responses of guinea pigs to specific synthetic polymorphic antigens were controlled by a single gene: the immune response gene (Ir). He pointed out that immune phenomena are governed by this gene. Their research made great contributions to immunology and was rewarded the Nobel Prize in Physiology or Medicine in 1980 [6].

## 3. Structural Characteristics of MHC Gene

Based on experimental findings, it has been determined that the mouse MHC gene is situated on chromosome 17 and comprises the K, I, S, D, and L regions. Within the I region, there are further subdivisions into subregions IA and IE [7]. In contrast, the human MHC gene is located on chromosome 6 and encompasses the A, B, C, DP, DQ, DR, C4, Bf, and C2 regions [8,9] (Figure 1). The MHC gene exhibits three primary characteristics. (1) Diversity [10,11]: The structure of the MHC gene is intricate, with diversity primarily encompassing polygenicity and polymorphism [12]. Polygenicity refers to a gene complex consisting of numerous closely adjacent gene loci [13], and its encoded gene products also have the same or similar functions. Interestingly, the presence of pseudo-genes and non-classical genes contributes to the augmentation of gene diversity. Polymorphism refers to the coexistence of numerous alleles on the same gene locus as a result of gene mutation or gene recombination [14]. The occurrence of MHC gene polymorphism may be closely related to harmful gene mutations. Consequently, the polygenicity and polymorphism of the MHC gene greatly increase the diversity of MHC. This diversity in MHC genes confers upon species a robust capacity to adapt to their environment. It ensures that the population is capable of producing an appropriate immune response when confronted with various pathogens [15]. (2) Haploid inheritance: MHC genes situated on the same chromosome exhibit minimal recombination within the homologous chromosome, resulting in the formation of a closely linked gene cluster, which is known as haplotype or haploid. This haplotype is inherited as a single unit by the offspring, which is termed haplotype inheritance [16]. The MHC genotype is composed of two homologous haploids. Consequently, the offspring inherits one haploid type from the male parent and another haploid type from the female parent, both of which are identical to those of their respective parents. This implies that one allele exhibits dominance in heterozygotes, thereby suggesting the capacity of an individual to amplify the expression of MHC molecules upon the expression of each gene locus (Figure 2). (3) Linkage unbalance: This occurs because specific alleles from distinct MHC loci consistently co-occur in inheritance. The linkages between genes are not entirely random, with certain genes consistently being linked together. Consequently, this leads to an increased occurrence of certain genes, compared to a scenario in which both genes were randomly and simultaneously present [17,18]. Thus, MHC genes have unique structural features that are common in biological evolution. Roger et al. evaluated linkage unbalance as it affects the HLA-B gene, among numerous renal transplant candidates originating from southern Brazil. The findings revealed a prevalence of linkage unbalance among haplotypes [19]. Consequently, the identification of a MHC gene linkage imbalance can serve as a valuable attribute when elucidating and scrutinizing the evolutionary patterns within a population [18,20].

## 4. Classification, Structure, and Function of MHC

The genes encoding MHC are situated on segments of the same chromosome and form a closely linked cluster of genes [21]. The MHC gene family produces various glycoproteins. Based on their distinct structural and functional characteristics, MHC genes can be classified into three separate gene subfamilies: class I, class II, and class III [22,23,24,25]. MHC class I and class II serve as executors for the purpose of antigen presentation, while MHC class III functions as either a complement component or a tumor necrosis factor [26]. The products generated by MHC I genes are typically located on the cell surface and are responsible for monitoring intracellular conditions. For example, when cells are invaded by viruses, MHC I plays a crucial role in recognizing the amino acid chains in the viral outer membrane fragments. Subsequently, these fragments are presented to CD8^+^ T lymphocytes, triggering a series of immune responses aimed at eliminating the virus [27]. On the other hand, the majority of MHC II molecules are located in antigen-presenting cells (APCs) [28], including macrophages, dendritic cells (DCs), B cells, and Langerhans cells (LCs) [29,30,31]. These MHC II molecules primarily serve to monitor and gather information about the extracellular environment. For instance, when bacteria invade tissues, macrophages are capable of engulfing and degrading them. Subsequently, MHC II molecules present bacterial antigens to CD4^+^ T lymphocytes, thereby assisting both cellular and humoral immune responses [32]. The encoding of complement components, tumor necrosis factor, heat shock proteins, and other such molecules is primarily attributed to the MHC III gene. In contrast to the first two classes of MHC, MHC III exhibits a lower level of polymorphism, resulting in comparatively limited research on this class [33,34].

The structure of MHC molecules is intricate. MHC I molecules are heterodimers composed of a heavy chain (α chain) and a light chain (β chain) [35]. The β chain is also referred to as β2-microglobulin. The α chain and β2-microglobulin are non-covalently bound together [36]. The α chain consists of three external structural domains, including α1, α2, and α3 [37,38]. Each of these domains is composed of approximately 90–100 amino acids. Following the α3 domain is a transmembrane domain that comprises 25 hydrophobic amino acids. The α1 and α2 domains interact to form the antigen-binding groove in MHC I molecules [38]. As the antigen-binding groove is closed at both ends, possible antigen binding to MHC I molecules is limited in length. Most of them are short peptides with a length of 8–11 amino acids [39]. The β chain is positioned centrally within MHC I molecules and plays a crucial role in stabilizing the molecular structure [40]. The β2-microglobulin structural domain exhibits a similar size and structure to the α3 domain, and interacts with the α chain of MHC I molecules non-covalently. The β2-microglobulins lack transmembrane regions and are encoded by distinct genes that display a high degree of conservation (Figure 3).

The structure of MHC II molecules is similar to that of MHC I molecules, which also comprises two polypeptide chains, α chain, and β chain [41]. Each chain of MHC II molecules consists of two extracellular domains. The α chain includes α1 and α2 structural domains, and the β chain includes the β1 and β2 structural domains. For MHC II molecules, the antigen-binding groove is formed by α1 and β1. In contrast to the MHC I molecule, the distinctive feature is that both ends of the antigen-binding groove are open. The length of this open structure determines its ability to accommodate peptides consisting of 10–18 amino acids [42] (Figure 3).

## 5. Antigen Presentation of MHC Molecules

Antigen presentation is a crucial process for the body to identify and respond to antigens. This process allows for differentiation between endogenous and exogenous molecules. The subsequent recognition and response to foreign molecules serve to protect the body from pathogens and other detrimental agents. The immune response generally encompasses three fundamental processes: antigen recognition and presentation, T cell proliferation and activation, and antigen clearance [43]. Notably, MHC molecules, APCs [44], and T cells [45] are important components of the immune system. Therefore, APCs play a crucial role in various tasks preceding the activation of T cells, encompassing antigen internalization, processing, and presentation [46]. This responsibility is inherently determined by the distinctive expression of MHC molecules in APCs [47,48]. Subsequently, specific representative APCs will be examined in further detail.

MHC I molecules are almost expressed in all nucleated cells [49]. As the primary antigen presentation molecules, they present immunogenic antigens on the cell surface and activate immune response. Specifically, viral antigens or tumor antigens expressed in the cell are referred to as endogenous antigens [50]. These endogenous antigens are usually degraded by the proteasome into short peptides consisting of 8–11 amino acids [51]. Subsequently, these short peptides are transported to the endoplasmic reticulum (ER) via the antigen processing and presentation transporter (TAP) [52]. Therein, they bind to the MHC I molecules and form antigen peptide-MHC complexes, which are subsequently transported to the cell surface. These complexes then interact with CD8^+^ T lymphocytes and convert them into cytotoxic T lymphocytes (CTLs). This activation initiates cell-mediated immunity and the targeted killing of infected cells. This pathway exemplifies the conventional presentation mechanism of MHC class I molecules [53] (Figure 4).

The functions of MHC II molecules are similar to those of MHC I, while MHC II molecules mainly present exogenous antigens [54]. Specifically, protein antigens from inactivated vaccines or subunit vaccines [55], as well as particulate antigens from pathogens, are considered exogenous antigens [56]. The exogenous antigens are usually internalized by APCs and create endosomes or phagosomes. These endosomes and phagosomes are able to fuse with the MHC II compartment (MIIC) in the cytosol [28]. The MIIC is rich in various enzymes and acidic milieu that account for the degradation of exogenous antigens into short peptides consisting of 13–18 amino acids [57]. These short peptides bind to MHC II and form stable peptide-MHC II complexes, which are then transported to the cell surface. These complexes interact with the receptor on CD4^+^ T lymphocytes, forming trimolecular complexes [58], thereby activating CD4^+^ T lymphocytes and initiating an immune response [59] (Figure 5).

## 6. Cross-Presentation of MHC Molecules

In the classical presentation pathway, MHC I molecules mainly present endogenous antigens and activate cellular immunity. However, when APCs present exogenous antigens to MHC I molecules, CD8^+^ T cells are also activated to initiate cellular immunity. This process is referred to as cross-presentation [60,61]. MHC I molecules have the ability to present exogenous antigens derived from degraded bacteria or parasites on the cell surface. These antigens are recognized and bound by the TCR [62,63], resulting in the activation of CD8^+^ T lymphocytes and subsequent initiation of a cellular immune response against target cells [64]. MHC I molecules deliver exogenous antigens through two main pathways: the cytosolic pathway (also known as the TAP-dependent pathway) and the vesicular pathway (also known as the TAP-independent pathway) [65]. Both pathways of antigen presentation involve three essential steps: the degradation of antigens into short peptides [66], the formation of MHC I short peptide complexes [67], and the transportation of these complexes to the cell surface [68].

### 6.1. Cytosolic Pathway

The cytosolic pathway involves the transporting of exogenous antigens from the endocytosis to the cytosol [69]. During this process, professional APCs, particularly DCs, transport exogenous antigens from the endocytic vesicles to the cytosol. Within the cytosol, the exogenous antigens are digested by proteasomes, resulting in the generation of small peptide fragments containing 8–11 amino acids [70]. These peptides are then transported by TAP to the ER, where they bind to MHC I molecules. After undergoing processing and modification in the Golgi apparatus, they are transported to the cell surface [71].

Research has revealed that APCs possess a limited capacity to internalize antigens upon recognition. Consequently, a substantial concentration of antigens is necessary for cross-presentation to take place [72]. This is why small dose injection of exogenous antigens, such as vaccines into the body, is insufficient to induce CD8^+^ T lymphocyte immune response. Thus, the enhancement of exogenous antigen internalization by APCs is crucial for augmenting cross-presentation. Furthermore, phagocytosis serves as the initial stage in the internalization process of exogenous antigens. This process involves the formation of phagosomes through the fusion and recombination of endocytic vesicles, lysosomes, and ER-related structures. Within the cytosolic pathway, phagosomes may potentially fulfill a protective role for antigens. The study indicated that DCs possess the ability to elevate the pH value in phagosomes, thereby facilitating antigen cross-presentation and inhibiting antigen hydrolysis [73]. In alkaline environments, antigens are less prone to degradation, facilitating the participation of more antigens in the cytosolic pathway [74]. Meanwhile, this reaction can cause lipid peroxidation, which leads to the rupture of the phagosome, releasing antigens into the cytosol and augmenting cross-presentation. Previous studies have shown that mouse bone marrow dendritic cells (BMDCs) recruit NADPH oxidase 2 (NOX2) for ROS reactions, but the deletion of NOX2 inhibits antigen cross-presentation, although not completely [75,76,77]. This suggests the existence of alternative mechanisms that facilitate the release of antigens from the phagosome into the cytosol.

The cytosolic pathway of antigen presentation is reliant on TAP transporters, while certain antigens also rely on endoplasmic reticulum aminopeptidase I (ERAP1) with shearing activity [78], and ERAP1 functions by cleaving exogenous antigens into shorter peptides prior to their delivery to the ER. Research has indicated that several components of the ER, including TAP, Sec61, and Sec22b, can be recruited and introduced into the phagosome in a specific manner [79]. Furthermore, Sec61 exhibits a reverse transport mechanism that transports peptides back into phagosomes, thereby co-mediating the degradation of exogenous antigens to generate peptide recycling [80,81]. Through retroactive transportation, the peptides will be further hydrolyzed by phagosomal proteases. One of these proteases is called insulin-regulated aminopeptidase (IRAP) [82]. Similar to ERAP, IRAP possesses the ability to remove most amino acids from the N-terminus of the peptides, thereby trimming peptides into short peptides for adaption to the MHC I molecules.

The protein molecules involved in the cytosolic pathway are essentially identical to those in the classical presentation pathway [83]. The membrane compartmentalized structure along the phagosomal membrane can retrogradely transport peptides that have been degraded by proteasomes. These peptides are transported back into the phagosome through the TAP, which has been recruited from ERGIC. IRAP plays a role in trimming these peptide segments into a suitable form for presentation by MHC I molecules, following the antigen transportation to CD8^+^ T lymphocytes. Consequently, it constitutes a complete transport pathway to deliver antigen peptides to the cell surface (Figure 6).

### 6.2. Vesicular Pathway

In the process of antigen cross-presentation, the vesicular pathway cannot rely on TAP transporter protein. It is capable of generating ligands for MHC I molecules through the endocytosis pathway and assembling them within endocytosed vesicles. Peptides presented via this pathway are produced by direct degradation of exogenous antigens in phagosomes in the presence of cysteine protease inhibitors [84]. Notably, Cathepsin S within the acidic lysosomes plays a crucial role [85]. Currently, it is widely believed that the vesicular pathway is less effective than the cytosolic pathway [86].

Antigen processing for cross-presentation via the vesicular pathway takes place in endocytosed vesicles. Initially, it was thought that the MHC I molecules were located in the near-nuclear region [87]. However, decades of experimental studies have shown that cross-presentation is more likely to occur when antigens interact with exosomes or phagosomes [88]. Peptides and MHC I molecules constitute a crucial part of cross-presentation. Therefore, understanding the source of MHC I molecules loaded with exogenous antigen peptides is particularly important for uncovering the molecular mechanisms of cross-presentation. It has been found that MHC I molecules equipped with antigenic peptides are also recycled [89]. However, the mechanism of endocytosis and recycling of MHC I molecules in the vacuolar pathway has not been systematically understood (Figure 7). The investigation of the intracellular distribution of MHC I molecules within the vesicular transport system necessitates further research in the field of cross-presentation.

Clayton and Spel et al. made important discoveries in the model of viral infection and cross-presentation of soluble ovalbumin [90,91]. They demonstrated that conserved tyrosines in the intracellular tails of MHC I are required for their endocytosis and their targeting of lysosomal vesicles. The antigenic peptide presented by MHC I in this pathway is likely to be produced in the endocytic pathway. MHC I tyrosine residues in the cytosolic tail contribute to endocytosis. However, the molecular mechanisms of lysosomal targeting and cytosolic adapters that recognize MHC I transport motifs have not been elucidated.

## 7. Cells Performing Cross-Presentation

### 7.1. Dendritic Cells

Antigen cross-presentation is not only performed in professional APCs, but also in some special cells [92]. Therefore, DCs are considered the most efficient cells for antigen cross-presentation. DCs are usually categorized into two groups: conventional DCs (cDCs) and plasmacitoid DCs (pDCs). cDCs are further divided into two major subsets: cDC1s and cDC2s [93]. The development of cDC1s is dependent on the transcription factors IRF8 and BATF3 [94], while cDC2s only require the transcription factor IRF4 [95]. cDC1s are the most functionally prominent cells for antigen cross-presentation. For instance, soluble chicken ovalbumin can be presented by mouse CD8^+^ cDC1s in lymphoid tissues [96], and cross-presented by CD103^+^ cDC1s in non-lymphoid tissues, including intestines [97], thymus, and skin [98]. Moreover, human cDC2s also support cross-presentation to exogenous antigens, which function like cDC1s and pDCs [99,100].

### 7.2. Liver Sinusoidal Endothelial Cells

Liver sinusoidal endothelial cells (LSECs) constitute an abundant non-parenchymal cell population in the liver, playing crucial physiological and immunological roles, including the ability to cross-present soluble antigens [101]. Schurich et al. [102] analyzed the kinetic properties of cross-presentation by LSECs and DCs, and found that LSECs significantly exhibit higher efficiency in capturing circulating antigens from the blood compared with DCs, macrophages in the spleen, or other liver cell populations. In vitro, quantitative studies of antigen concentrations revealed that LSECs had superior uptake and cross-presentation of antigens compared with DCs [103]. To some extent, the ability of LSECs to cross-present soluble antigens in vivo is as prominent as that of dedicated APCs.

### 7.3. T Cells

As an integral component of the innate immune response, γδ T cells possess specific TCRs that are encoded by the Vγ- and Vδ-genes. In the peripheral blood of the normal population, the proportion of γδ T cells typically ranges from 2–10% [104,105]. These TCRs have the ability to recognize nonpeptide compounds derived from bacterial or tumor cells, a process that does not involve the participation of MHC molecules [106]. However, research has found that human Vγ9Vδ2^+^ T cells increase the expression of MHC II and CD86, which are characteristic markers of APCs, in response to plasmodium falciparum infection [107]. Furthermore, it has been demonstrated that Vγ9Vδ2^+^ T cells exhibit the functional characteristics of professional APCs, which present antigens to CD4^+^ and CD8^+^ T cells in vitro [108]. Therefore, γδ T cells may possess the ability to conduct cross-presentation.

## 8. Transporting of MHC Molecules

MHC molecules are an integral part of cross-presentation; however, the source of the MHC molecules and the mechanism of their transportation in the cytoplasm are not well defined. Currently, it is generally accepted that MHC molecules can be transported in three ways: trogocytosis, exosome secretion, and tunneling nanotubes [109].

Trogocytosis refers to vesicular transportation between living cells, which involves intercellular uptake of specific cell surface proteins [110]. The process of trogocytosis involves the dynamic transference of vesicles among live cells, instead of phagocytosis of apoptotic vesicles [111,112]. Similar to other immune cells, DCs may acquire membrane vesicles and intracellular proteins from other cells by trogocytosis [113]. Notably, trogocytosis is not a protein-selective transfer process. Therefore, transmembrane proteins and MHC molecules are generally passively involved in the vesicular transfer of trogocytosis.

Exosomes are membranous vesicles with a diameter ranging from 50–100 nm [114]. They enter the cell through the invagination of the endosomal membrane and are released into the extracellular environment [115]. Exosome secretion has been reported in a variety of cell types [116], including mast cells, DCs, reticulocytes, platelets, tumor cells [117], B lymphocytes, intestinal epithelial cells, T lymphocytes, and microglia [118]. Exosomes can transport certain substances that are captured by MHC molecules [119]. Wakim and Bevan [120] demonstrated that DCs can utilize exosomes to transport endogenous MHC I-like molecules loading captured peptides. Thus, exosomes may serve as an effective way to facilitate MHC transportation in DCs.

Tunneling nanotubules (TNT) is a long membrane protrusion process that facilitates the exchange of cell surface molecules and cytosolic content between cells [121]. Experimental data have shown that TNT can mediate the transportation of MHC I-like molecules and allow cross-presentation between remote DCs [122]. Meanwhile, this exchange of membrane molecules can be a unidirectional phenomenon between epithelial cells and DCs [123] or a bidirectional phenomenon between DCs [124].

In addition, the efficiency of MHC molecule transportation is dependent on the expression capacity of MHC proteins in donor cells [125]. It has been shown that when mouse dendritic cells are co-cultured with allogeneic DCs or endothelial cells, the efficiency of peptide-MHC I complexes transport is lower when compared to that for MHC II complexes [126], on account of the higher abundance of MHC II molecules compared with MHC I molecules in DCs.

## 9. Future Perspectives

MHC molecules perform a wide range of biological functions, including participating in tissue and organ transplantation, vaccine immunization, inoculation, and autoimmune diseases. The most crucial biological role of MHC molecules lies in mediating antigen presentation in the immune response. Compared with a classical presentation, cross-presentation is of great significance in immunity, because a large and diverse spectrum of antigens is generated during cross-presentation. Moreover, in many pathological and physiological processes, cross-presentation is thought to be a major pathway for the activation of CD8^+^ T lymphocytes to initiate an immune response. However, due to a lack of evidence, it is not yet possible to accurately evaluate the effects of cross-presentation in immune response and disease. Despite there being ongoing progress towards advancing the understanding of cross-presentation, many new questions continue to arise. For example, what are the mechanisms of the recycling and translocation of MHC I molecules? The role of MHC I in immune tolerance is not yet fully understood, and some key links in the antigen presentation pathway are still in question. Meanwhile, the use of the cross-presentation pathway for disease treatment is still in its early stages in terms of clinical practice. In-depth studies of its mechanism of action will be helpful to ensure its more efficient application in the future.

## Figures and Tables

**Figure 1 ijms-25-00196-f001:**
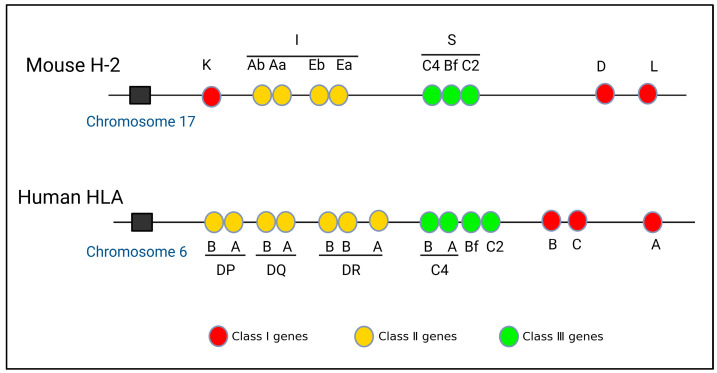
Gene structures of mouse H-2 and human HLA.

**Figure 2 ijms-25-00196-f002:**
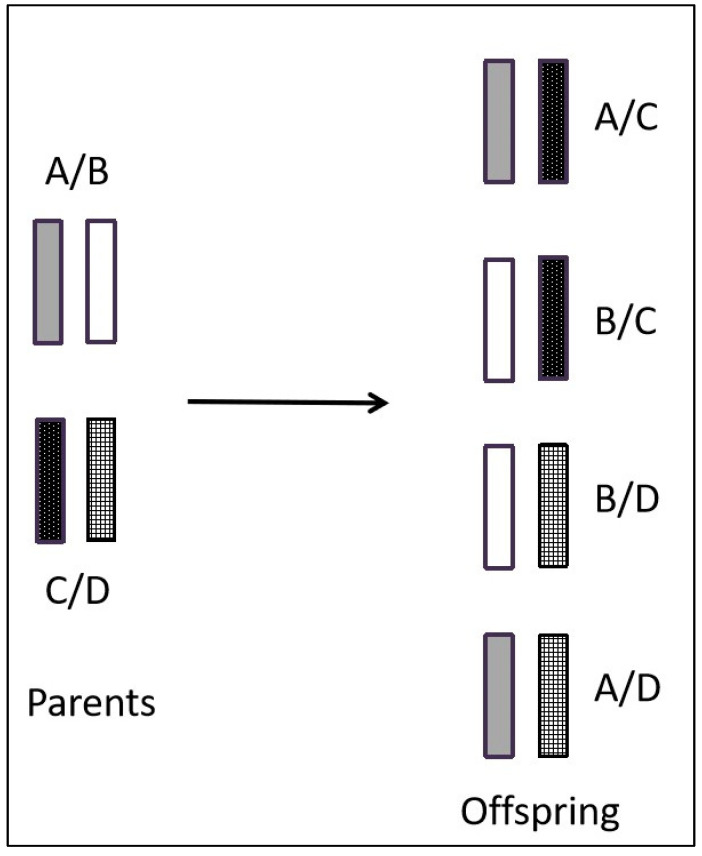
Pattern of haplotype inheritance of MHC genes.

**Figure 3 ijms-25-00196-f003:**
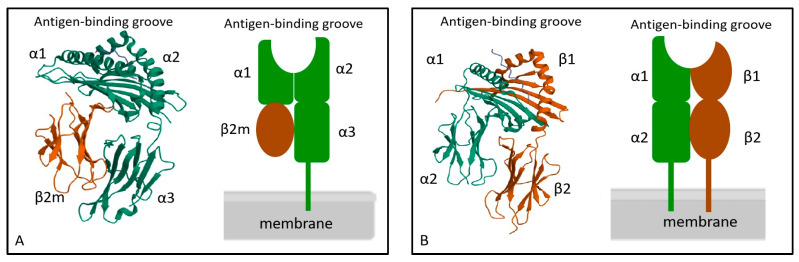
Molecular structure of MHC I (**A**) and MHC II (**B**). MHC I molecule is composed of α chain and β2- microglobulin, α1 and α2 constitute the antigen-binding groove of MHC I. The MHC I molecule is a heterodimer composed of a transmembrane α chain and a non-transmembrane β2- macroglobulin. MHC II molecule is composed of α chain and β chain, α1 and β1 constitute the antigen-binding groove of MHC II. The MHC II molecule is a heterodimer composed of two transmembrane α and β chains.

**Figure 4 ijms-25-00196-f004:**
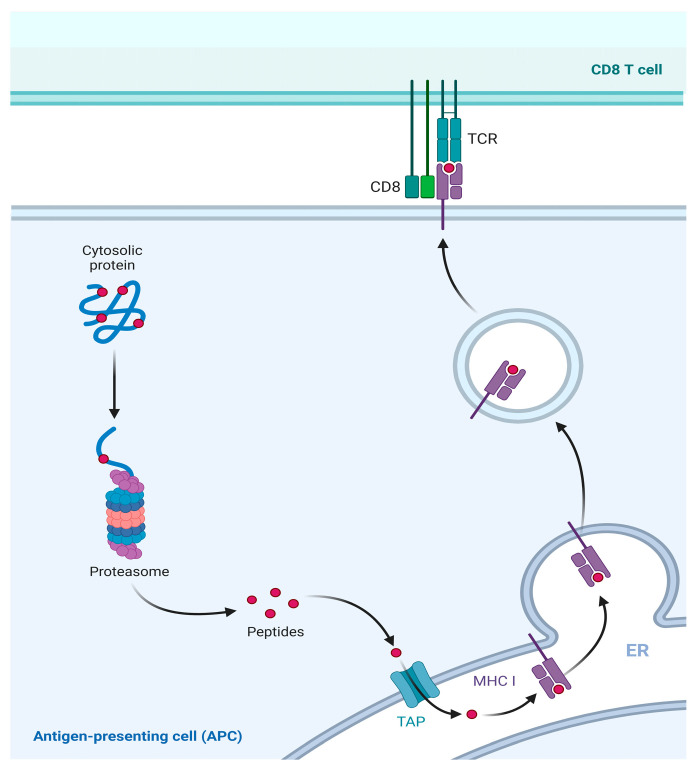
Classical presentation pathway of MHC I molecule. MHC I molecule presents the endogenous antigens. Intracellular expressed viral or tumor antigens are degraded into short peptides by proteasomes in the cytoplasm. Then, short peptides are transported into the ER through TAP. Therein, peptides bind to MHC I to form a complex and to be presented to the cell surface for recognition by CD8^+^ T cells.

**Figure 5 ijms-25-00196-f005:**
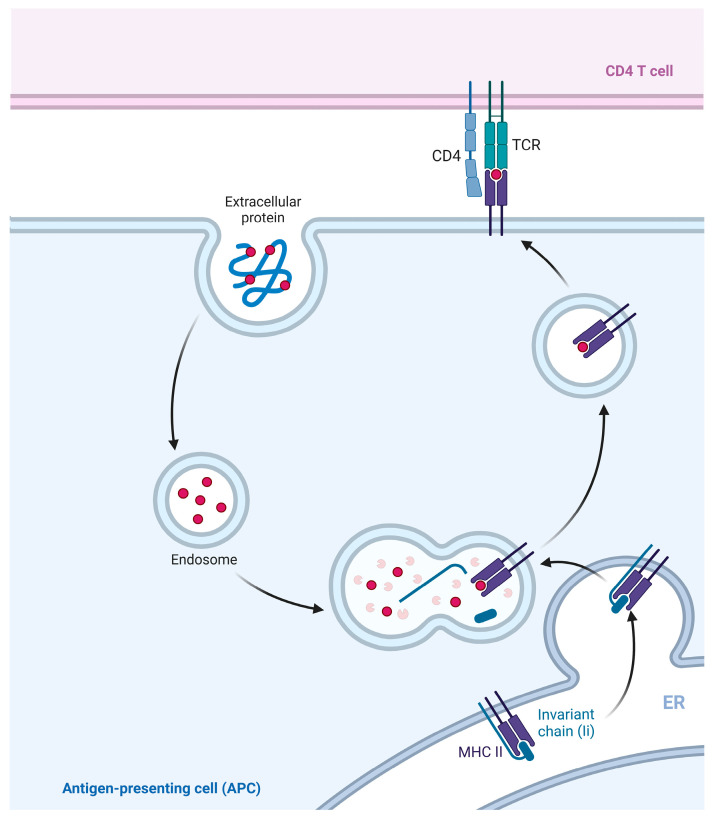
Classical presentation pathway of MHC II. Exogenous antigen was recognized and extracted by APCs in the formation of intracellular phagosomes. (MHC II/li) 3 nontamers synthesized and assembled in the ER are involved in MIIC formation. The phagosome fused with MIIC, and the exogenous antigen was degraded in MIIC. Chain Ii is also degraded in MIIC, leaving the CLIP in the antigen-binding groove of MHC II. CLIP is displaced with antigenic peptides to form peptide-MHC II complexes, which are then transported to the cell surface.

**Figure 6 ijms-25-00196-f006:**
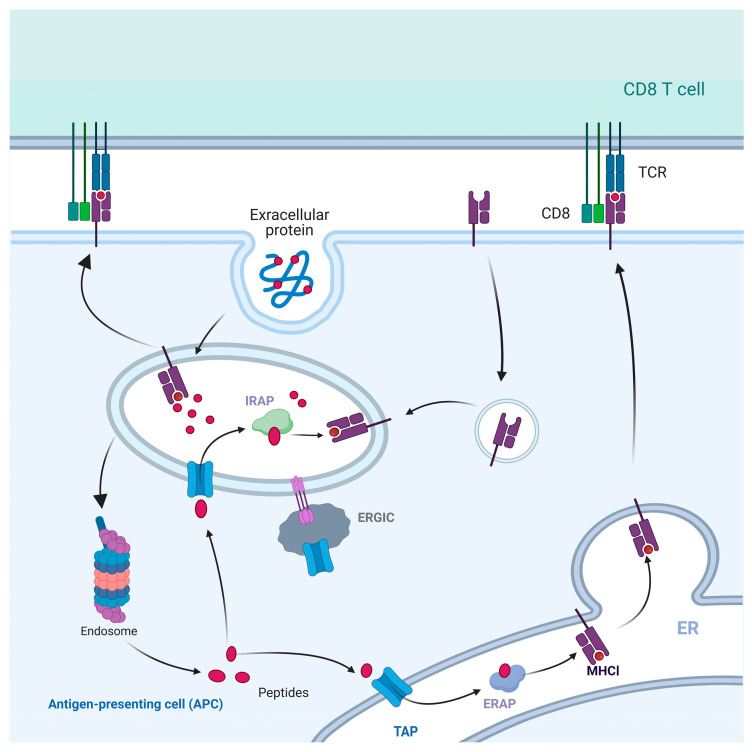
Cytosolic pathway of cross-presentation. Exogenous antigens are phagocytosed by phagosomes and short peptides are degraded by proteasomes. These short peptides are transported to the ER by TAP, and cleaved by ERAP to become mature peptides for binding to MHC I. Peptide-MHC I complexes are presented to the cell surface as ligands to the TCR. Some peptides degraded by proteasome are transported back to the phagosome through TAP and cleaved by IRAP directly, then bind with MHC I molecules in the phagosome and transported to the cell surface.

**Figure 7 ijms-25-00196-f007:**
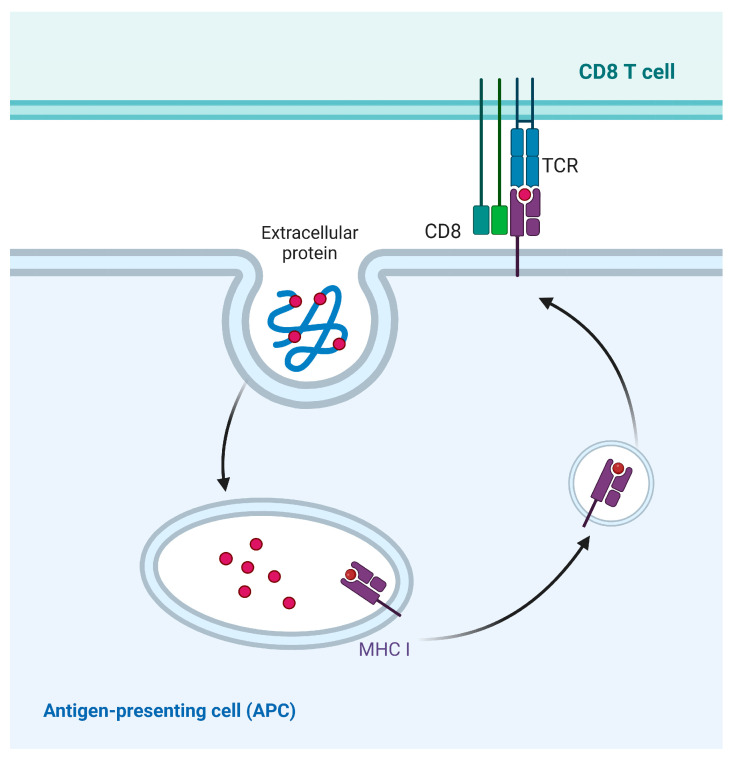
Vesicular pathway of cross-presentation. Exogenous antigens are phagocytosed and degraded by phagosomes, and are directly transported to the cell surface by phagosomes, where peptides are loaded to MHC I molecule.

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
