# Peer review of "Molecular Characteristics, Functional Definitions, and Regulatory Mechanisms for Cross-Presentation Mediated by the Major Histocompatibility Complex: A Comprehensive Review"

_ijms, 2023, doi:10.3390/ijms25010196_

Round 1
Reviewer 1 Report
Comments and Suggestions for Authors
Remarks to the Authors:
In this manuscript, the authors summarized the MHC molecular function, structure and the mechanism of cross-presentation. That’s interesting, but there are still some suggestions to improve this manuscript.
1. In the figure 2, there are some Chinese words in the picture, I think this journal is English editing, so the authors should pay more attention in this minor mistakes. Dr. Liu, you are a researcher, please don’t make this mistake again, though this a review, you should check this manuscript for minor mistake before submitting.
2. The figure 2 & 3 should be combined, and the more details should be provided for the more informative of this picture.
3. Also, for the figure, the image and letters of Figure are not clear. This figure also needs to be reorganized to mark up the key results. Moreover, the authors should provide detailed information for this Figure in the Results section.
4. In the title, the authors marked this manuscript is a comprehensive review, however, authors just simply listed the results for each part. For example, in the section of Cells performing cross-presentation, there is no discussion I can find, authors should increase content in every content.
5. The English writing should be enhanced.
Comments on the Quality of English Language
English writing should some improvements.
Author Response
Thank you very much for receiving your reply. All my responses to you are in the file.
Please see the attachment.

Reviewer 2 Report
Comments and Suggestions for Authors
Unfortunately, some part of the review is incomprehensible There are also some important oversights. For example: one figure it’s written in Chinese, some references are absent (lane 70 and lane 165) or old (lane 74). The description of MHC I (lane 87-9) is incorrect, regions and domains are confused, and the PM of the beta chain is missing. In some parts, it is not clear; we don’t understand if we are talking about MHCI or MHCII (lanes 152 -158).
Comments on the Quality of English LanguageEnglish is not clear and in some cases some words are placed random.
Author Response

(The authors gave the same response as above.)

Round 2
Reviewer 1 Report
Comments and Suggestions for Authors
Remarks to authors:
I don't have anymore question. I just want remind you that please be serious, obviously, you have made the same mistake again.
Reviewer 2 Report
Comments and Suggestions for Authors
The review has improved, but still needs some changes.
Lane 49: ‘L’ is missing from the mouse MHC gene transcript. It would also be important to add a reference.
Figue 1: Add MOUSE (H-2) and HUMAN (HLA)
Lane 155: extracellular domain
Figure 3: Add the legend to the figure for example ‘A/B’ or ‘MHC I/ MCII’
Figure 3: molecular structures are the same in both MHC I and MHC II
Lane 144: almost in all or perhaps all nucleated cells?
Lane 182-192: perhaps it would be better to move this part to the beginning of the paragraph
Lane 203/209/210: cytosolic pathway
Lane 276-277: Initially, it was thought that the MHC I molecules were in the proximal nuclear region. What do the authors mean?
Lane 302-314: Rewrite this very important paragraph better, paying attention to the classification of dendritics and add review to lane 305 (for example, plasmacitoid DCs and not plasma cell- DCs, CD 103+ what are they, cDC1s, cDC2s or pDCs…? )
I hope I have been helpful in improving your review.
Round 3
Reviewer 2 Report
Comments and Suggestions for Authors
The review has improved considerably.
Good luck with you work.